# Automated Quality Assessment of Interferometric Ring Laser Data

**DOI:** 10.3390/s21103425

**Published:** 2021-05-14

**Authors:** Andreas Brotzer, Felix Bernauer, Karl Ulrich Schreiber, Joachim Wassermann, Heiner Igel

**Affiliations:** 1Department of Earth and Environmental Sciences, Ludwig Maximilian University Munich, Theresienstr. 41, D-80333 Munich, Germany; fbernauer@geophysik.uni-muenchen.de (F.B.); jowa@geophysik.uni-muenchen.de (J.W.); igel@geophysik.uni-muenchen.de (H.I.); 2Geodetic Observatory Wettzell, Research Unit Satellite Geodesy, Technical University of Munich, 93444 Bad Koetzting, Germany; ulrich.schreiber@tum.de

**Keywords:** rotational seismology, rotation rate sensors, ring laser data processing, ROMY, data quality control, Sagnac effect

## Abstract

In seismology, an increased effort to observe all 12 degrees of freedom of seismic ground motion by complementing translational ground motion observations with measurements of strain and rotational motions could be witnessed in recent decades, aiming at an enhanced probing and understanding of Earth and other planetary bodies. The evolution of optical instrumentation, in particular large-scale ring laser installations, such as G-ring and ROMY (ROtational Motion in seismologY), and their geoscientific application have contributed significantly to the emergence of this scientific field. The currently most advanced, large-scale ring laser array is ROMY, which is unprecedented in scale and design. As a heterolithic structure, ROMY’s ring laser components are subject to optical frequency drifts. Such Sagnac interferometers require new considerations and approaches concerning data acquisition, processing and quality assessment, compared to conventional, mechanical instrumentation. We present an automated approach to assess the data quality and the performance of a ring laser, based on characteristics of the interferometric Sagnac signal. The developed scheme is applied to ROMY data to detect compromised operation states and assign quality flags. When ROMY’s database becomes publicly accessible, this assessment will be employed to provide a quality control feature for data requests.

## 1. Introduction

In seismology, rotational motion observations supplement the conventional observation of translational motions and extend the scope of insight granted by seismic observations (e.g., [1,2,3,4]). Array derived rotations, based on classical translational seismometer array measurements (e.g., [5,6,7]) and recent efforts to derive rotational motions using high-rate GNSS measurements [8], were complemented by rotational sensors in the course of the last decades (e.g., [9,10]). Stationary ring laser installations, such as G-ring at the Geodetic Observatory in Wettzell, Germany (operating since 2002) and the four-component ring laser array ROMY at the geophysical observatory in Fürstenfeldbruck, Germany (operating since 2017), represent highly sensitive optical instruments in terms of absolute rotations. Ring lasers are optical interferometers [11] and exploit the Sagnac effect [12,13]. For large ring laser installations with a cavity length exceeding four meters, we distinguish between monolithic and heterolithic structures. While the data quality of monolithic instruments greatly benefits from its inherently high mechanical stability, heterolithic devices are subject to a larger sensor drift, due to a lower mechanical stability. A monolithic design, commonly made of the class ceramic Zerodur, is technically limited in cavity length (< 16 m). With each ring having a cavity length of about 36 m, ROMY represents a heterolithic structure made of stainless steel and concrete. Hence, in the absence of an active cavity stabilization system, we are dealing with a drift of the optical frequency in the interferometer. Since the free spectral range FSR=c/P, with *c* being the speed of light and *P* the perimeter, is around 8.6 MHz for ROMY’s rings, the frequent changes of the longitudinal mode index of the oscillating laser modes result in two often observed scenarios:(1)a loss of contrast as the currently oscillating dominant laser mode is weakening, while it drifts away from the maximum of the laser gain curve and(2)a sudden change of laser modes, where the two counter-propagating laser beams are not operating on the same longitudinal mode index.

In the second scenario, we loose the interferogram entirely from our detector bandwidth, since the interferogram is biased away from the usually used audio frequency range by multiples of the FSR. In order to evaluate the ring laser output automatically, we need a reliable criterion to evaluate and flag the validity of the detected signal for all components of ROMY in near real-time. Such a data flag does not only exclude erroneous signals from the observation record, it also provides the necessary feedback signal to recover the proper operation state of each ring laser.

This paper is structured as follows. We first review the theoretical foundations of the Sagnac principle as used in ring laser technology. This is followed by a description of the data processing scheme developed for the quality assessment of the raw ring laser signal. Finally, the scheme is applied to real multicomponent data recorded by the ROMY ring laser, followed by discussion and conclusions.

## 2. Fundamental Background on Ring Laser Instruments

### 2.1. Sagnac Interferometry

The Sagnac effect describes the effect of rotation on the propagation of light around a closed contour. In the case of an external light source, this effect shows up as a phase shift between the co- and the counter-rotating light beam [12]. If, however, we use laser excitation in a ring resonator, this effect causes a frequency shift, which allows us to observe a beat note δf, when interfering the counter-propagating laser beams [14] (see Figure 1C,D). The beat note frequency depends on the scale factor of the Sagnac interferometer, more precisely the ratio of the enclosed area *A* and the perimeter *P* of the resonator. δf is strictly proportional to the rate of rotation experienced by the entire apparatus and can be expressed as:(1)$$\delta f=\frac{4A}{\lambda P}\;\vec{n}\;\vec{\Omega }\;,$$
where λ denotes the laser wavelength and $\vec{n}$ the normal vector on the area of the sensor. A ring laser, which is rigidly attached to the solid Earth, thus observes the Earth rotation vector $\vec{\Omega }$ as the primary signal.

It is important to note that the Sagnac effect is an effect on the propagation of light, which allows for the observation of the motion of the encapsulating sensor housing (cavity) filled with a low pressure gas mix relative to the two counter-propagating light beams on a closed, reciprocal path [11]. Some of the properties of an optical Sagnac interferometer that come along with specific requirements are as follows:an entirely linear transfer function up to the Nyquist frequency as long as the sensor does not deform under external forces (e.g., centrifugal forces).a well-resolved study of Earth’s rotation with merely one large-scale, ground-based sensor as long as the ring cavity is strapped down to the rigid Earth and shows stable long-term performance in terms of the optical frequency as well as the cavity geometry, which poses a great challenge.a high-sensitivity observation of seismically induced ground rotation commonly below 10^−7^ rad/s, which are contained in the measurement as perturbations of the rather uniform beat note of Earth rotation [15,16]. Detecting these signals requires a large and sensitive Sagnac interferometer and the ROMY ring laser array represents such a sensitive device.

Rotational rates are obtained by sophisticated processing with an instantaneous frequency estimation at its core, as outlined by Igel [17]. Rather than evaluating the quality of the rotation rate time series, we present an approach to classify the quality based on the raw beat note signal and thus infer important information on the instruments operation state at the same time.

### 2.2. Characteristics of a Large-Scale Ring Laser Array

ROMY abbreviates **RO**tational **M**otions in seismolog**Y** and represents the only existing four component large-scale ring laser array, unprecedented in its design and scale [17,18]. It was constructed in 2016 at the site of the Geophysical Observatory in Fürstenfeldbruck, Germany. The setup and working principle is schematically shown in Figure 1. Each component is realized as an equilateral, triangular ring laser with approximately 12 m side length and arranged to form a downward pointing tetrahedron. The evacuated resonant cavity of each ring is filled with a helium-neon gas mix. Laser functions are established with radio frequency excitation of a plasma at the gain tube (see Figure 1B). Three low-loss super mirrors form a closed optical beam path. In order to make the instrument highly sensitive [11], the cavity is designed for minimal loss.

The fundamental limit of an optical gyroscope is the shot noise of the photon flux on the detector. Since this has the characteristics of white noise, it averages down according to: $1/\sqrt{t}$, with *t* being the observation time. Practically more important is the Q-factor of the ring laser cavity, which determines the linewidth of the laser radiation and can be quantified by measuring the ringdown time. The ringdown time of a cavity depends on its losses, which vary considerably across ROMY’s rings at this point in time. We observe that the horizontal ring is about 1.5 orders of magnitude more sensitive than the worst of the slanted rings [17]. A possible cause is dust contamination of the bottom mirrors in the slanted cavities, which can thus be remedied in principle. ROMY has a large heterolithic optical cavity made from stainless steel, referenced to a solid concrete foundation. Temperature variations, mechanical vibrations and ongoing settling of the newly built monument cause a substantial drift of the optical frequency in the cavity. While most of these perturbations are a common mode effect and cancel out in the interference, some of it does not cancel, since the laser process is not entirely reciprocal. Backscatter coupling and a nullshift offset are responsible for variable drift effects in the interferogram, which affect the long-term stability of the sensor. As a result, ROMY does not yet resolve variations of Earth rotation. In the short-term, however, ROMY performs very well given that the two counter-propagating laser modes are lasing on the same longitudinal mode index. If that is the case, we can typically observe a slow variation of the nominal beat note at the level of 10 ppm over several hours, which is an effect caused by gain medium dispersion and backscatter coupling [11,19]. A nominal Sagnac frequency characterizes each of ROMY’s equilateral triangles being defined by the projection of Earth rotation onto its respective normal vector. A difference of observed and expected nominal Sagnac frequency is caused by a small misalignment towards north and a small inclination to the horizontal [18]. At the moment, all four components achieve a sensor resolution of the order of 10^−11^ rad/s, as shown by the Allan deviation for integration times between 60 and 200 s [17]. A large-scale ring laser as ROMY has the potential to reach a resolution below 10^−13^ rad/s, if the laser cavity was fully stabilized, thus enabling one to routinely observe not only solid Earth tides, but also the Chandler Wobble. Moreover, an array of ring laser is capable to resolve the complete Earth rotation vector due to its four components, as demonstrated for ROMY [18].

### 2.3. Ring Laser Operation in the Presence of Optical Frequency Drift

The laser transition of the helium-neon ring laser is about 1.6 GHz wide and we use an equal mix of ${}^{20}$Ne and ${}^{22}$Ne isotopes to decouple the two beams from each other. Lasing is achieved near the line center, where the gain is at a maximum. We employ gain starvation in order to reduce the number of excited laser modes in the cavity, thereby running the system at very low gain. Intra-cavity mode selection devices are prohibitive, since they would reduce the Q-factor and hence the resolution. Due to the massive size of the cavity of 36 m, it only takes a cavity variation by one wavelength to cause a frequency drift of about 8.6 MHz, thus a higher mode activation. In the presence of thermal expansion, this would cause rapid transitions of longitudinal modes. This effect is reduced by overpressuring the cavity, causing a suppression of neighboring laser modes over a range of about ±100 MHz by homogeneous line broadening. Longitudinal mode changes nonetheless occur frequently and given the narrow FSR we have the difficulty that both laser beams match in longitudinal mode index after such a mode transition. Although this does not remove the Sagnac effect from the measurement, it pushes the interferogram outside the sampling window of the digitizer and therefore becomes undetectable. Furthermore, the set of laser modes becomes unstable when it faces such a mode transition, due to mode competition effects. During such a transition phase, the mode contrast is dramatically reduced and as a consequence the interferogram is not stable. In the absence of a perimeter stabilization mechanism, we have to detect and exclude the periods of compromised operation states and the corresponding observations from the data analysis procedure. Moreover, a detection of the presence of different mode indices allows one to recover proper operation by automatically triggering a restart of the lasing process.

## 3. Methodology

### 3.1. Evaluation Scheme: Quality Measures

The developed evaluation scheme is designed to detect operation states compromised by mode competition effects or the loss of an interferogram due to the presence modes with different longitudinal mode index. On the basis of this scheme, a quality flag is assigned to the individual data sections. In order to judge not only the quality of the seismological important rotation rate but also why and how the signal is distorted, the introduced scheme is based on analysis of the raw sinusoidal signal obtained by Sagnac interferometry. Due to nominal frequencies of up to about 553 Hz (z-component), the beat note signal is sampled at a rate of 5 kHz with 24 bit resolution, which results in a large data volume. In order to automatically calculate quality measures, 15 min chunks of raw data (=4.5 × 10^6^ samples) are queried and the quality measures, as defined below, are calculated for 2 s sliding windows with a 50% overlap. No significant improvement was found for windows below 2 s. The obtained values on these 2 s windows are either averaged or the corresponding extreme values are extracted for 20 s intervals, which correspond to the sampling rate of the quality measures (= quality samples). For the assessment of individual data sections, the following signal characteristics or quality measures are determined:**Mean value:** calculated mean values (M) of 2 s windows are averaged for 20 s intervals. The resulting function relates to an averaged laser intensity level, which is amplified by the photo detector. It is used to monitor automatic restarts of the lasing process and ensure lasing is initiated. Additionally, $d_{t}M$ is computed as the absolute of a first-order finite difference of the mean value (*M*) to enhance the identification of jumps in laser intensity.**Frequency estimate:** represents a simple count of zero-crossings over 2 s windows. The median of the frequency ($f_{\mathrm{sagnac}}$) over 20 s serves as a quality measure. A deviation of the specific, single-mode nominal frequency of each ring indicates the presence of multiple longitudinal mode indices.**Maximal amplitude**$A_{\max}$ and **minimal amplitude**
$A_{\min}$ for each 2 s window are computed and the median over 20 s is used to monitor the stability of the maximal peak-to-peak amplitude ($A_{pp}$) of the Sagnac signal.**Signal contrast Γ**: is computed according to: $\Gamma =\left(A_{max}-A_{min}\right)/\left(A_{max}+A_{min}\right)$ and $\partial_{t}\Gamma$ represents its derivative based on a first-order difference. The contrast ranges from 0 to 1, were a high contrast reflects a high-quality interferogram at low laser intensity, while a low-contrast signal indicates a potential mode transition due to optical frequency drift.**$\Delta A_{\mathrm{ext}}$** is computed as the maximal variation of peak-to-peak amplitudes in a sub-window. $A_{pp}$ is calculated for each 2 s window of data. Across each 20 s interval, which corresponds to the sampling rate of the quality measures, the maximum and minimum of the set of $A_{pp}$ values is determined. The difference of these maximal and minimal values is stored as $\Delta A_{ext}$. This serves as a measure of the maximal variability of the peak-to-peak amplitude over an interval of 20 s. This provides a measure to detect variations of the interference signal caused by mode competition effects due to a developing frequency drift.

Figure 2 shows three signal cases: (1) desired single-mode, high-contrast signal, (2) intensity increase causing a low contrast and (3) distorted signal due to an appearing second longitudinal mode. For each 2 s bin, the signal characteristics are indicated, excluding the frequency estimate. For illustration purposes, the shown amplitude variations are exaggerated and the nominal frequency is reduced.

### 3.2. Evaluation Scheme: Thresholds

In order to classify detections of compromised operation states and assign a quality flag for each 20 s interval, three quality levels are defined:**Q0:** good quality data with high contrast (=green).**Q1:** medium quality data, allowing fluctuations in the signal contrast (=yellow).**Q2:** bad quality data, not suitable to be used for processing and analysis (=red).

The thresholds needed for the assignment of the corresponding quality levels are inferred from an empirical analysis of existing data of ROMY. The most relevant signal characteristic is the nominal frequency of the corresponding ring, which is the projected Earth rotation in single-mode operation. An interval of ±1.5 Hz (approx. 1.5 × 10^−7^ rads/s) centered around the respective nominal frequency ($f_{Z}$ = 553.4 Hz, $f_{U}$ = 303.0 Hz, $f_{V}$ = 447.5 Hz, $f_{W}$ = 451.7 Hz) is defined as a quality criteria for the frequency estimate. Compromised operation states cause the nominal frequency to exceed this frequency interval and are not suitable for the calculation of rotation rates, thus assigned Q2 quality. In order to act more robust, an additional criterion is used to also classify single quality samples of Q0 or Q1 surrounded by samples of Q2 as Q2 quality data. This helps one to avoid false assessment due to frequency estimates during operating phases of unstable frequency (Figure 3C). The feedback controlled beam intensity is kept at a low level to facilitate gain starvation of higher lasing modes and corresponds to a voltage between 0.5 and 1.5V. For the mean values (*M*), a minimal threshold of 0.1 V is set to avoid false positives for unpowered rings due to electronic noise randomly meeting Q0 or Q1 criteria. Exceeding an upper threshold of 2 V results in an assignment of Q1, since it significantly reduces the contrast and indicates mode competition effects. Sudden intensity changes are detected by $d_{t}M$ with a threshold set to 0.02 V/s and often coincide with an automatically triggered reset of the lasing process. The calculated contrast Γ monitors the sinusoidal amplitude of the beat note signal with respect to the overall intensity. A high and stable contrast is linked to a good interference at single-mode operation. As part of manual maintenance the contrast is commonly optimized by varying the gain of the photo detectors and therefore no nominal contrast is applicable as a threshold. However, a minimal contrast of 0.08 is inferred, and lower levels are classified as Q2. Exceeding the threshold defined for $\Delta A_{ext}$ by 0.3 V is another criteria that results in an assignment of Q1 quality. This threshold aims at the detection of periods with fluctuations of the peak-to-peak signal, which is related to mode competition effects.

## 4. Results and Discussion

For validation purposes, a data selection of 14 November 2019, is shown in panel A and B of Figure 3, comprising the raw beat note signal and the obtained rotation rate with distorted sections of high amplitude, respectively. Several examples show the effects of optical frequency drift and their detection. Panels C to H of Figure 3 display the corresponding quality measures and the applied thresholds. Between minute 566 and minute 583 of Figure 3, the intensity increases to a level of 3 to 4 V, decreasing the contrast and causing several quality measure to pass their thresholds. Accordingly, the samples are classified as Q1 and Q2, as shown color-coded in Figure 3. Two examples of an operation on different longitudinal modes, starting at minute 613 and minute 625 (Figure 3), respectively, are indicated by a disappearing interference signal (panel A and F) and detected by $f_{sagnac}$, $\Delta_{ext}$, Γ and $\partial_{t}\Gamma$. An interferogram could be restored after a mode competition had been triggered. When the presence of different longitudinal modes is detected, a control software automatically triggers a restart of the lasing process involving a brief boost of the power (M≈ 5 V) at gain tube of the cavity, resulting in the excitation of several lasing modes. The natural mode selections controlled by the characteristic gain curve of the resonator eliminates higher modes, thus ideally achieving a single-mode operation. Several successful, automatically triggered restarts can be seen in Figure 3 (t=616min,625min,656min). If re-establishing a single-mode operation is not successful, a manual maintenance is necessary. Depending on the situation, this can result in a longer downtime for the acquisition of good quality data. This sort of maintenance often includes a realignment of the lasing path and realignment of the radio frequency excitation, which varies the laser intensity. A detection of lost Sagnac interference is shown in Figure 3 at around t=616min. If the difference $\Delta A_{ext}$ exceeds a value of 0.3 V, Q1 quality is assigned. An example is provided in Figure 3 at *t* = 597 min, where fluctuations of the interferometric signal are detected based on this criterion. Figure 3 also shows that several measures are required to work together to avoid wrong classification or false positives.

Furthermore, the scheme was applied to data of all four components of ROMY for November 2019. During this period an active field experiment was conducted at the site of the observatory, for which ROMY served as a reference station and all four rings were operating. Panels A–D of Figure 4 display the daily share of all quality levels across November 2019 for each of ROMY’s four components Z, U, V and W, respectively. Apart from a data gap on 27 November and maintenance work between 11 November and 14 November, the z-component demonstrates the best performance in terms of percentage and continuity of Q0 data. Here, the slanted rings confirm an increased vulnerability to optical frequency drift effects and underline a currently missing long-term stability. Overall, Q1 quality accounts for only a small share. The median frequency estimates shown in Figure 4, based on Q0 and Q1 data only, remains stable within a tolerance of 0.25 Hz, while the frequency offset for the z-component is related to maintenance work. The corresponding contrast for the z-component included in Figure 5A, being on an overall low level (≈0.2), shows a further decrease before maintenance was required due to drift effects that could not be automatically recovered.

Figure 5 shows the evaluated data of ROMY’s z-component for the entire month of November 2019 not only by daily shares but also resolved temporarily across daily hours. Such a temporal visualization demonstrates the distribution of data sections with compromised operation states and subsequently the potential of a selection of a time series with good quality only. While several failures interrupt a stable performance of the instrument at the beginning of November, a continuous record of Q0 data is significantly improved for the second half of November.

An application of the scheme for six hours of ROMY’s w-component are shown in Figure 6, where rotation rates are shown with the quality levels as the background. Four occurrences of mode competition effects are detected (Q1), with the last resulting in an operation state where multiple longitudinal modes with different mode index are present (Q2). Additionally, rotation rates of a regional event in Albania on 28 November 2019 at 10:52:43 UTC with a magnitude of 4.7 are shown in the record.

The presented scheme will help to provide quality measures for ring laser data of ROMY and possible other instruments, when the database becomes publicly available. It is planned to run a service similar to the FDSN webservice “waveform catalogue” [20], which consists of a database with corresponding quality measures entries. The remote user will then be able to select only those data which meet predefined quality criteria. This could either be data without any gaps or data with Q0 quality only.

## 5. Conclusions

Realized as a heterolithic structure, ROMY’s four ring lasers experience an optical frequency drift as a consequence of mechanical instability of the resonant cavities. The large scale factor provides ROMY with unprecedented sensitivity to absolute rotations, however, at the same time reduces the free spectral range and facilitate longitudinal mode transitions and an operation of counter-propagating beams with different longitudinal mode index, both compromising a stable Sagnac beat note. An automated quality evaluation scheme for ring laser data has been developed, using the recorded raw Sagnac beat note signal to infer quality measures and classify the record into three quality levels. Using the raw interferometric signal allows to gain important information on the sensor’s current performance and health state. For this purpose, several quality measures, which jointly characterize the interferometric Sagnac signal, are computed. Empirically defined thresholds for these quality measures are applied to assign quality levels. The automated evaluation scheme is applied for data of November 2019 for all four components of the large-scale ring laser ROMY. This reveals multiple detection of mode transitions and amplitude fluctuations due to mode competition effects as well as compromised operation states due to counter-propagating beams with different longitudinal mode index in an automated way. The scheme will be employed to provide the remote user with an opportunity to request data according to quality levels as soon as the ROMY database becomes publicly available.

## Figures and Tables

**Figure 1 sensors-21-03425-f001:**
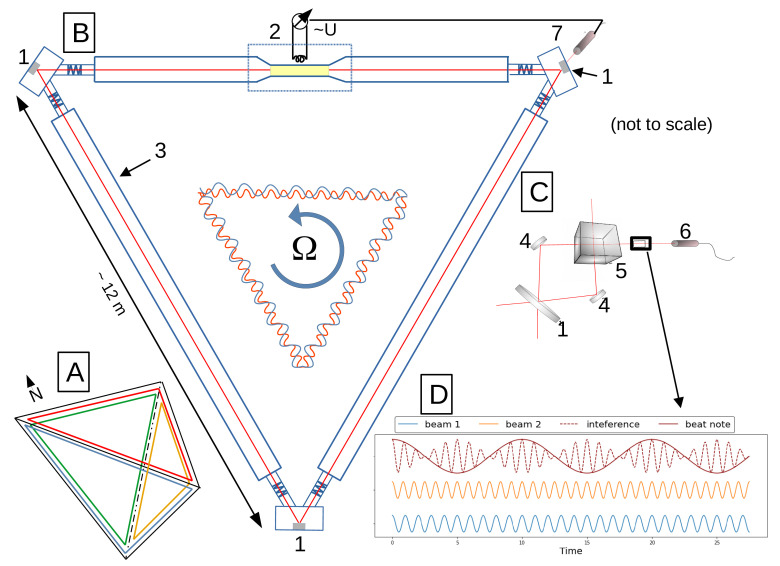
(**A**) all four components of ROMY form a downward pointing tetrahedron. (**B**) a schematic cutaway drawing of the design of one of ROMY’s components, each realized as an equilateral, triangular ring laser. (1) The triangular lasing path is closed using highly reflective mirrors. (2) gas discharge is sustained inductively with a radio antenna at the narrowed gain tube. (3) evacuated resonant cavity filled with a Helium-Neon mix. An illustration at the center shows the frequency shift between counter-propagating laser beams introduced by rotation Ω according to the Sagnac effect. (**C**) combination of the two beams using two deflection mirrors (4) and a beam combiner (5) is installed at the lower corner of the ring. A photo multiplier (7) records the monobeam intensity used to stabilize the laser intensity via a control loop.(**D**) illustration of the interference (dashed red) of counter-propagating laser beams (blue and orange) with shifted frequencies which create the beat note (solid red) that is recorded by a photo multiplier (6).

**Figure 2 sensors-21-03425-f002:**
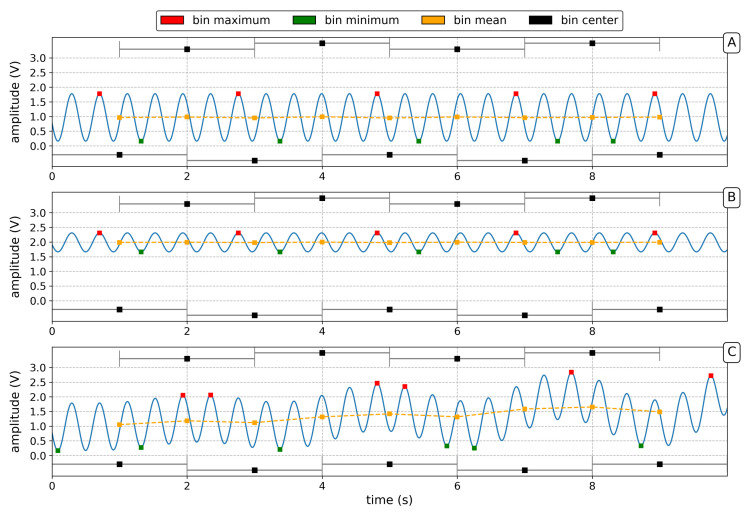
Schematic example of the quality measure computation for: (**A**) Single-mode, high-contrast signal; (**B**) A single-mode, low-contrast signal due to mode competition effects; (**C**) An appearing multimode operation state. Shown are synthetic signals for 2 s windows with exaggerated amplitude variations and reduced frequency for demonstration purposes. Mean (orange), maximal value (red) and minimal value (green) is calculated for each bin as well as an estimate of the frequency based on zero-crossings (not shown). Bins with 50% overlap are indicated (black).

**Figure 3 sensors-21-03425-f003:**
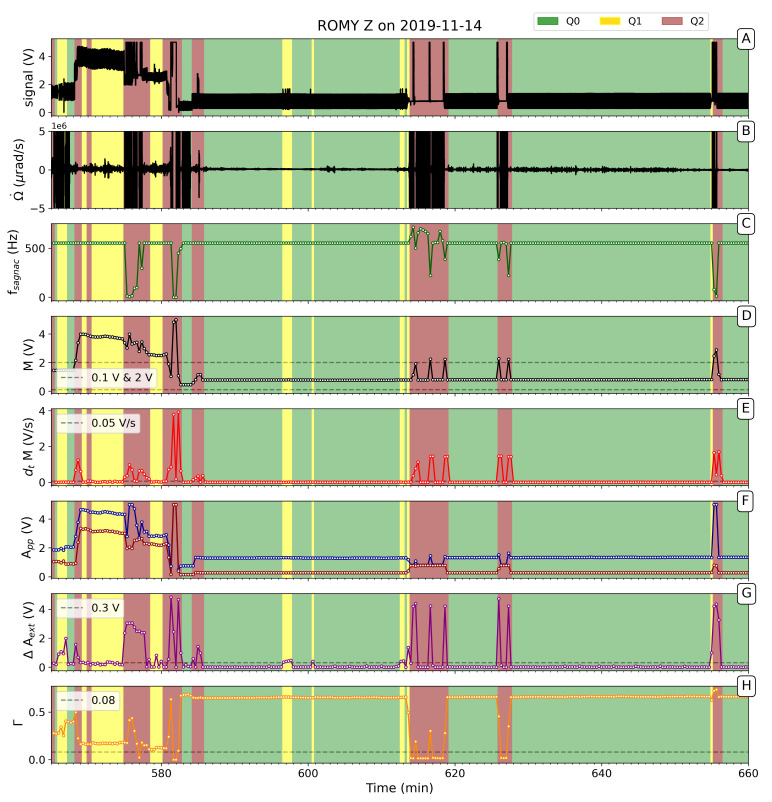
Selected data section of ROMY’s z-component on 2019-11-14. (**A**) Raw beat note signal; (**B**) Obtained rotation rate $\dot{\Omega }$ (clipped at ±5μrad/s); (**C**) Estimated Sagnac frequency; (**D**) Mean values *M*; (**E**) first-order finite-difference of *M*; (**F**) Median of maximal and minimal amplitudes $A_{pp}$; (**G**) Maximal variance of peak-to-peak amplitudes $\Delta A_{ext}$; (**H**) Calculated contrast value Γ. Quality evaluation thresholds are indicated as grey, dashed lines. The background colors correspond to the classified quality level.

**Figure 4 sensors-21-03425-f004:**
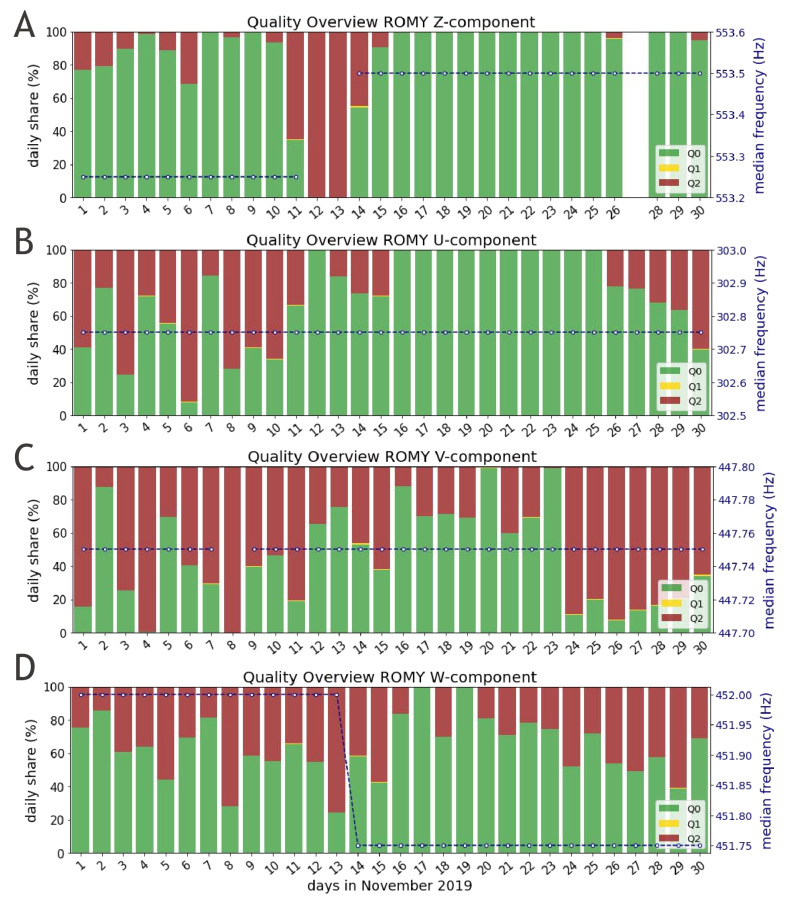
Quality assessment applied to all four components of ROMY (**A**–**D**) for November 2019. Quality measures are computed for 20 s intervals. Daily shares shown in green, yellow and red resemble the qualities Q0, Q1 and Q2, respectively. The daily median of frequency estimates, based on Q0 and Q1 data only, is plotted on a second y-axis (blue) for each component (**A**–**D**), respectively. No data is available for the z-component on 27 November 2019.

**Figure 5 sensors-21-03425-f005:**
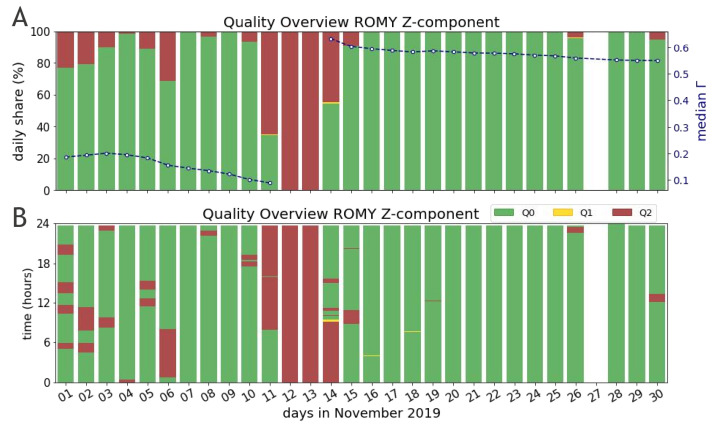
Data of ROMY’s z-component for November 2019 is shown in (**A**) as daily shares and in (**B**) across daily hours, in order to visualize the temporal quality distribution. No data is available for the z-component on 27 November 2019. The data is evaluated for 20 s based on 2 s windows and color code according to three quality levels Q0 (good), Q1 (medium) and Q2 (bad). (**A**) additionally shows the daily median of the contrast Γ computed by using Q0 and Q1 data only.

**Figure 6 sensors-21-03425-f006:**
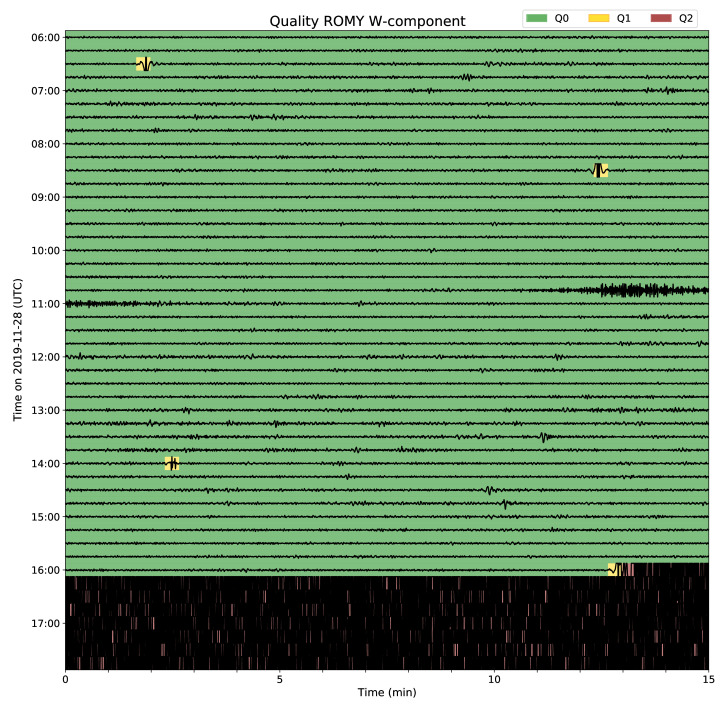
Six hours of data of ROMY’s w-component on 28 November 2019 are shown, including a regional event in Albania (Mw = 4.7) at 10:52:43 UTC. The rotation rate record is mean corrected, bandpass filtered (0.1 to 1.0 Hz) and clipped. Background colors indicate assigned quality levels for 20 s intervals.

## Data Availability

Second level data presented in this study are openly available in a publicly accessible repository: https://syncandshare.lrz.de/getlink/fi72mpaCFH1XbPnMwoDYarzY/. Raw data of ROMY are not yet publicly accessible due to the large amount of data, but available on request from the corresponding author.

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
