# Peer review of "Automated Quality Assessment of Interferometric Ring Laser Data"

_sensors, 2021, doi:10.3390/s21103425_

Round 1
Reviewer 1 Report
Andreas Brotzer et al
Automated Quality…..
The paper presents a method to make quality assessment of the Ring Laser gyroscopes (RLG) data. Taking into consideration the growing interest in angular rotation rate measurement for seismology, this paper is suitable for publication in this journal.
It is important to say that angular rotations of the Earth crust usually have very small amplitude, and instrumentation with sensitivity better than nrad/s is in principle necessary. RLGs are at present the only device sensitive enough and able to run in a almost continuous basis. Certainly the 10prad/s resolution obtainable with ROMY is out of reach for most instrumentation.
RLG has the drawback to exhibit changes of the operation point, producing for example spikes in the data, and split mode operation, which is basically useless when data are acquired with standard seismological instrumentation. To remove the bad portions of data is certainly wise. It is obvious that the geophysics has to concentrate in his analysis and has to use data already validated.
The authors use the fringe contrast of the interferogram to select the good dat portion, and they say that the common problems are loss of fringe contrast and mode competition of higher order modes. Synthetic signals are used to develop the method,it would have been better to use directly the real data. Thee level of data quality are elaborated: good, half good, bad. The method is used to asses the data quality of the ROMY data (November 2019). The fringe contrast is commonly used to select the good portion of data of RLG data.
The paper is well written, despite I would prefer a more simple and compact language in a scientific paper.
The title is declaring that the method is general, while in my opinion it is mainly for ROMY. Other heterolithic RLGs are already operative for seismology, but they are not taken into account by the author because the threshold of 16m2 is chosen. It is not clear to me why they pose that threshold, in practice suitable for seismology are all the devices with sensitivity better than nrad/s in 1 second measurement, and as far as I understand FOGs with sensitivity even lower are commonly used. The threshold, if effectively necessary, should be posed on sensitivity more than on size.
Fig. 1 provides a general picture of ROMY, the part D is trying to explain the interference, I appreciate the effort to provide and intuitive explanation of the beating of the two waves, but I believe that this pictorial view is not clear at all, and I propose to eliminate it from the figure. If I understand well the amplitude control of one of the monobeams is implemented, but in fig.1 it is not shown.
There are several statements regarding the performances of RLG and of ROMY which are not relevant for the paper (the focus being the quality assessment), the comprehension of the paper would improve concentrating the discussion around the focus. The general remarks should be reported in the introduction, concentrating the following discussion and information for the paper focus explanation. The introduction should also report the general situation of RLGs which have given relevant contribution to seismology.
There are several statements regarding the performances of RLG and of ROMY which are not relevant for the paper (the focus being the quality assessment), the comprehension of the paper would improve concentrating the discussion around the focus. The general remarks should be reported in the introduction, concentrating the following discussion and information for the paper focus explanation. The introduction should also report the general situation of RLGs which have given relevant contribution to seismology.
In the following I indicate the suggested changes:
line 68, eliminate this sentence, since it is not true. In fact it is true only when the non linearities of the laser are perfectly under control. The paper ignore completely the non linearities problem of the laser dynamic, and in my opinion this is correct, since the paper is devoted to geophysics, i.e. it should provide the basic elements to understand the RLG use, and the data quality. Accordingly it is correct to list the main problems, but not to say something which is not true.
Line 71, ….the Sagnac effect is due to the confrontation of the propagation of light in the two directions in a closed path…..,please remove relativistic to make it simpler
Line 73: eliminate ‘in a vacuum’. (The cavity is filled with gas)
Line 77: eliminate: under under centrifugal forces (the cavity deformates under the effect of any force, it is not required to be centrifugal….)
Line 84: This last point is useless, the only necessary information would be the level of sensitivity required to be meaningful for seismology. This concept is expressed several times, but never in a quantitative way. I suggest to decide the level of sensitivity required, before I have suggested the 1nrad/s level, but I may be wrong.
Line 116: eliminate the sentence: in practice this means……
Line 121: eliminate entire sentence: Backscatter…..
I have to say that after line 122 the description becomes confused and is not necessary for the comprehension of the paper focus. I suggest to eliminate it including in the introduction the main things related to ROMY which they would like to say.
Section 2.3 reports some important parameters of the functioning of the laser itself. The functioning of the laser is rather complex and the description is purely qualitative. I don’t know whether it is relevant to the comprehension of the paper. The only relevant part is the fact that they are able to automatically restart the lasing process in order to keep it running in the optimal mode.
It is always surprising for me to note that ROMY is operated without recording the two monobeams. Those two quantities are essential to understand the status of the laser, and they can be very easily monitored using very cheap common photodiodes, the use of more expensive devices as photomultipliers is not strictly required. The excitation of higher order modes should appear clearly in the monobeams, providing a very easy and clean way to identify and remove the related portion of data. My suggestion is to add photodiodes to the apparatus. The quality assessment method is not general, rather it is conceived for ROMY so the title should be more specific, I suggest to write ROMY -> of Interferometric Ring Laser
In short my suggestion is to concentrate in the introduction all comments related to ROMY, extending also the introduction citing the existing RLG suitable for seismology. The part describing the method should be more compact and concentrated on the things necessary to explain it: the list of the main problems and the method itself.
Author Response
Note: 1) Figure 1 has been slightly adopted and exchanged. 2) one reference Xu et al. 2019 has been added. 3) link to file share has been changed to different repository.
Please see the attachment for further point-to-point response.

Reviewer 2 Report
The manuscript is well prepared and well written. It is focused on the quality evaluation of measured data from ROMY. I know the German group has done a lot on instrumentations for rotational seismology. Thus a continuing report on their instrumentation and evaluation of data would help the reader of interest.
Here come my comments, which should be considered, before the manuscript can be published.
[1] The authors mention the problems of ROMY in Lines 40-43. They report an extremely high sampling rate for ROMY but only design some quality measures on a 2-second. The authors should probably explain why not a very sampling rate of data is used in their data quality evaluation;
[2] Figure 3 is not quite clear straightforwardly to me. Should this figure indicate that when motion exists, their corresponding data qualities become very poor, because the data corresponding to motion look like being marked in red color here in this figure. If my understanding is correct, why so? If not, please clarify;
[3] The authors should know that in addition to what has been mentioned in this manuscript, very precise and high-rate GNSS has been demonstrated to be of great potential to directly measure rotational motions due to large earthquakes. If baselines are sufficiently lengthy, Xu et al. (2019, Meas Sci Technol, 30, 024003) demonstrated this potential of GNSS rotational seismology, though rigorous theoretically but preliminary in terms of application since no GNSS has ever been designed to measure this motion. By properly designing a GNSS rotational measurement set, such a motion can be measured at the accuracy of 0.1 second to a few seconds, depending on the lengths of baselines. I believe that the authors should update the recent effort and progress on rotational seismology.
